# Socioeconomic status, depression, and incident arthritis in adults aged 50 years and over: Prospective evidence from six longitudinal aging studies

Lingyan Duan[1], Xiaomei Hu[1]*, Rui Qing[1], Rongyan Mou[1], Shuqi Chen[2], Zhu Lan[1], Fengming Zou[1]

1 The Second Affiliated Hospital of Zunyi Medical University, Zunyi, China, 2 Guangdong Provincial Hospital of Chinese Medicine, Guangdong, China

* gzhxm1988@126.com

## Abstract

### Objectives

To examine the independent and joint effects of socioeconomic status (SES) and depression with the risk of developing arthritis in adults aged 50+.

### Methods

This pooled study derived from six aging cohorts: the Health and Retirement Study in the United States (HRS), the Survey of Health, the Ageing and Retirement in Europe (SHARE), the English Longitudinal Study of Ageing (ELSA), the Korean Longitudinal Study of Aging (KLoSA), the China Health and Retirement Longitudinal Study (CHARLS), and the Mexican Health and Aging Study (MHAS). Cox proportional hazards models were applied to estimate independent and joint effects.

### Results

Over a median follow-up of 5.9 years among 68,445 individuals, 14,476 newly developed arthritis. After adjusting for covariates, low level of education (low vs. high: HR = 1.25, 95%CI: 1.19, 1.32), wealth (first quartile vs. fourth quartile: HR = 1.23, 95%CI: 1.17, 1.29), SES (low vs. high: HR = 1.36, 95%CI: 1.26, 1.47) and depression (HR 1.37, 95% CI: 1.32, 1.42) were independently associated with increased risk of developing arthritis. Individuals with depression consistently exhibited the greater risk for incident arthritis across education, wealth, or SES levels. Within each depression level, higher socioeconomic position was associated with decreased risk. A dose-response analysis revealed a steady increase in hazard ratios (HRs) as depression scores escalated, underlining significant positive relationships.

**Data availability statement:** All relevant data are within the paper and its Supporting information files.

**Funding:** This work was supported by the Joint Science and Technology Funding Projects by Zunyi Bureau of Industry and Science and Technology and the Second Affiliated Hospital of Zunyi Medical University (HZ(2024)427).The funder had no role in the design and conduct of the study; collection, management, analysis, and interpretation of the data; preparation, review, or approval of the manuscript; and decision to submit the manuscript for publication. The funders had no role in study design, data collection and analysis, decision to publish, or preparation of the manuscript.

**Competing interests:** The authors have declared that no competing interests exist.

## Conclusions

This multinational study presents the significant interactions of SES and depression on the incidence of arthritis in middle-aged and older adults, underscoring the need for targeted preventive measures and healthcare policies to reduce the disease's burden.

## Introduction

Arthritis is a chronic inflammatory condition, encompassing the major subtypes of osteoarthritis (OA) and rheumatoid arthritis (RA), and constitutes one of the most common causes of disability worldwide, especially among adults in mid and later life [1]. With rapid population aging, its prevalence continues to rise, now affecting more than one-fifth of individuals aged 50 years and older in large countries such as the United States, Canada, and China [2–4]. Beyond progressive joint damage, pain, and reduced mobility, arthritis can also compromise cardiovascular, renal, and neurological health, impose substantial financial strain, diminish quality of life, and contribute to excess mortality [5–8]. Given its growing recognition as a critical public health challenge in the context of global aging, arthritis poses a substantial obstacle to achieving healthy aging goals. Thus, identifying its risk factors and crafting targeted preventive measures are crucial steps in lessening its detrimental impacts on individual well-being and healthcare systems worldwide.

Socioeconomic disadvantage and depression both likely increase the risk of arthritis onset through multiple mechanisms. Lower education and limited wealth are linked to physically demanding work, adverse living conditions, unhealthy behaviors, and delayed preventive care, while depression impairs self-management and activity, disrupts sleep, and activates stress, endocrine, and immune pathways that sustain low-grade inflammation, together heightening susceptibility to incident arthritis [9–12]. Despite this consistent evidence for more active disease, whether socioeconomic status (SES) and depression predispose to the onset of arthritis has not been clarified. Only two case–control studies have evaluated SES in relation to incident arthritis [13,14], and evidence for depression largely comes from single-center studies [15,16]. Prospective, multi-country investigations examining socioeconomic status (SES) and depression as risk factors for incident arthritis in aging populations are lacking. This gap restricts our understanding of the temporality of these relationships and their generalizability across diverse global settings. Moreover, while the independent effects of SES and depression on the onset of arthritis have been extensively studied, their cumulative impact on arthritis development remain largely unexplored. Investigating this interplay is crucial for a deeper, mechanistic understanding of arthritis pathogenesis, which could reveal synergistic or joint effects not apparent when examining these factors in isolation.

This study seeks to fill these gaps by investigating the independent and joint effects of SES and depression on new-onset arthritis among adults aged 50 years and over across six studies. By integrating a broad geographical scope and a

longitudinal approach, this research aims to provide a nuanced understanding of how these factors interact to influence arthritis risk.

## Methods

### Study design and participants

In this pooled multi-cohort study, we utilized data from six prospective cohort studies across six studies in the Program on Global Aging, Health, and Policy [17].The Health and Retirement Study in the United States (HRS) is sponsored by the National Institute on Aging (grant number NIA U01AG009740) to the University of Michigan [18]. The Survey of Health, the Ageing and Retirement in Europe (SHARE) The SHARE study is subject to continuous ethics review by the Ethics Council of the Max Planck Society. [19]. The English Longitudinal Study of Ageing (ELSA) is founded by the National Institute of Aging (grants R01AG017644) and a consortium of UK Government departments coordinated by the Office for National Statistics [20]. The KLoSA has been approved by Statistics Korea (approval number: 33602) [21]. The China Health and Retirement Longitudinal Study (CHARLS) was approved by the Ethics Committee of Peking University (IRB approval number IRB00001052–11015) [22]. The Mexican Health and Aging Study is partly sponsored by the National Institutes of Health and National Institute on Aging (grant number NIH R01AG018016) [23]. All analyses used de-identified, publicly available data provided by these studies; no new data collection was conducted. To ensure a consistent timeframe for analysis across six studies, we analysed data from waves 10–14 of the HRS (2010–19), waves 6–9 of ELSA (2012–19), waves 3–7 of KLoSA (2010–18), waves 3–5 of MHAS (2012–18), waves 4–7 of SHARE (2010–18), and waves 1–4 of CHARLS (2011–18). The initial wave served as the baseline, while subsequent follow-up surveys were used to follow the outcome until the final wave. This study restricted adults aged 50 years and older, and participants who were missing information on exposure and outcomes, or had been diagnosed with arthritis at baseline were excluded. The selection process was presented in the flowchart (S1 Fig in S1 File).

All cohort studies have been approved by the relevant local research ethics committees. This is a secondary data analysis project and so specific ethical approval was not needed.

### Socioeconomic status

We utilized education and wealth to construct socioeconomic status (SES) in line with prior research [24,25]. Education levels were already harmonized into three categories: less than upper secondary (low), upper secondary and vocational training (medium), tertiary (high). Total household wealth was calculated as the net sum of all financial assets and was measured in the local currency of each country. We categorized total household wealth into quartiles within each country to ensure comparability, with Q1 to Q4 representing the lowest to the highest quartile. Detailed descriptions of education and total household wealth were shown in the S14 Table in S1 File.

Education and wealth might reflect different components of SES and thus might not be interchangeable [26]. Consistent with prior research [24,25], we adopted two approaches to measuring SES: (1) a summed score approach, combining the numerical values of education level (1–3) and household wealth quartiles (1–4), with total scores ranging from 2 to 7, and (2) a categorical approach, merging education and wealth categories into 12 distinct groups (3 education levels × 4 wealth quartiles). For the summed score of SES, lower scores indicated a lower SES. To facilitate analyses, we further classified the summed SES scores into four categories: low SES (score of 2), lower-middle SES (scores 3–4), higher-middle SES (scores 5–6), and high SES (score of 7).

### Depression

Probable depression, defined as having a depression symptom score above established cut points, was assessed using shortened versions of the Center for Epidemiological Studies Depression (CES-D) scale and the EURO-D scale. Previous

studies have applied both measurements within the same research [27], and detailed information on the scales and their validation among older adults has been published elsewhere [28–31].

Both scales ask about recent presence of negative affect (e.g., sadness) and somatic symptoms (e.g., trouble sleeping), with the CES-D additionally inquiring about positive affect in the past week and the EURO-D timeframe being the previous month. Scores were constructed by assigning 1 point for each negative affect/somatic symptom present and 0 if absent, with positive affect items reverse-coded. We used the 8-item CES-D for the USA and England surveys; the 9-item CES-D for the Mexican surveys; the 10-item CES-D for China and Korea; and the 12-item EURO-D for studies in Austria, Belgium, Czech Republic, Denmark, Estonia, France, Germany, Greece, Hungary, Italy, Netherlands, Poland, Portugal, Slovenia, Spain, Sweden, and Switzerland. The condensed versions of the CES-D and EURO-D, despite utilizing different time frames (1 week for CES-D vs. 1 month for EURO-D) and containing several various items, have been demonstrated to possess similar and comparable classification properties [27,32].

In this study, we adhered to previously established recommended thresholds [27], classifying participants with scores of ≥3 for the 8-item CES-D, ≥4 for the 9-item or 10-item CES-D, and ≥4 for the EURO-D as having probable depression. While these measures are not intended to diagnose clinical depression according to conventional diagnostic criteria, for the sake of simplicity, individuals scoring above these thresholds are referred to as experiencing "depression."

## Covariates

Participants reported age at baseline and gender. For consistency among multiple countries, body mass index (BMI) was calculated from height and weight ($kg/m^2$), and classified into underweight, normal, overweight and obesity based on country-specific cut-offs. Marital status was classified as either married/partnered or other (including separated, divorced, never married, or widowed). Smoking status was differentiated into never smokers and ever smokers, with the latter encompassing both former and current smokers. Similarly, drinking status was divided into never drinkers and ever drinkers. The level of physical activity was classified into two categories: engaging in moderate or vigorous activities at least once a week, and less than once a week. However, for the MHAS cohort, due to its specific categorization, physical activity levels were classified as "3 or more times a week" or "less than 3 times a week," thus using "3 times/week" instead. High blood pressure, diabetes, stroke, cancer, and lung disease were defined as a self-reported physician diagnosis of the conditions. "Whether or not a doctor has ever told the respondent they had hypertension or high blood pressure", and response "yes" was regarded as "had high blood pressure". This applies to other diseases as well. Detailed descriptions of covariates were shown in the S14 Table in S1 File.

## Outcome ascertainment

The primary outcome for this study was incident arthritis, which was ascertained based on self-reported physician-diagnosed. Participants were inquired if they had been informed by a doctor about having arthritis or rheumatism, either in the past or presently. Subjects who had arthritis at baseline were excluded, and if the patient was diagnosed with arthritis any time after that and up to the follow-up period, he or she was considered to have new-onset arthritis. The endpoint of follow-up was at the first occurrence of arthritis, death, or the censoring date, whichever came first. The censoring date was the date of the last survey each participant attended within their respective cohort.

## Statistical analysis

Baseline characteristics by arthritis were presented as frequencies with percentages for categorical variables and mean with standard deviations (SD) for continuous variables. Missing data of covariates were inputted via the R package 'miss-Forest' [33].

We analysed the associations between SES, depression and new-onset arthritis using Cox proportional hazards models stratified by countries, with follow-up time as the underlying timescale. Strata for countries in constructing models was used to address concerns about differences not fully accounted for by the available covariates and departures from the proportional hazard assumption [34]. We employed three models that progressively adjusted for various covariates. Mode 1 was a crude model with a strata term accounting for heterogeneity at the country level. Model 2 adjusted for age at baseline, and gender. Model 3 further adjusted for body mass index, marital status, smoking, drinking, physical activity, prevalent hypertension, diabetes, stroke, cancer and lung disease. We reported unadjusted and adjusted associations as the hazard ratios (HRs) and corresponding 95% confidence intervals (CIs). We assessed the independent associations of education level and total household wealth with arthritis incidence, followed by an analysis of the relationship between categorized summed scores of SES and arthritis. Additionally, we examined how various combinations of education levels and total household wealth were associated with arthritis risk.

Three types of analyses were performed using stratified Cox models to examine the joint effects of SES and depression on incident arthritis. First, we examined both the additive and multiplicative interactions between SES and depression in relation to incident arthritis. Second, stratified analyses were performed, where restricted cubic splines of changes in depression across SES categories were applied to graphically estimate the association of their combined effects. To ensure comparability across cohorts with depression scales varying from 8 to 12 items, we standardized the score. Then, an adjustment was made by adding the absolute value of the minimum score, ensuring the minimum value started at zero. Furthermore, the impacts of SES on incident arthritis were investigated within each level of depression. Third, the joint associations were subsequently investigated by splitting the overall sample into eight mutually exclusive groups based on levels of SES and categories of depression. We evaluated interactions between SES and depression using likelihood ratio tests comparing models with and without a cross-product term.

In supplementary analysis, we investigated joint effects of education, wealth, and SES alongside depression on arthritis incidence, stratified by age and gender. Additionally, we assessed associations between different combinations of education and wealth, and depression on incident arthritis, and these associations stratified by age and gender.

A series of sensitivity analyses were conducted. First, we performed sensitivity analyses by repeating models on sub-samples, removing one cohort at a time. Second, to examine the robustness of our results, three approaches were used to compare effects when accounting for the heterogeneity of study countries in the pooled dataset: country as a random term in random effects model, country as a covariate in multivariable regression model, and country as a stratum in stratified regression model (reference). Third, to address the issue of missing data, we reran the Stratified Cox regression models, treating all missing values as a separate category in the analysis.

All statistical analyses were conducted using R software version 4.3.1. All statistical tests were two-sided, and we considered a P value of less than 0.05 to be statistically significant.

## Results

The baseline characteristics of the included participants were presented in Table 1. Over the course of a median follow-up duration of 5.9 years among the 68,445 participants, 14,476 individuals were newly diagnosed with arthritis. The average age of the study population was 64.3 years (SD 9.7), and females constituted 51.1% of the participants. Compared to the non-arthritis group, individuals with arthritis were more likely to be older, female, obese, unmarried, and less physically active. They also tended to have lower levels of education and wealth, along with a higher prevalence of depression, hypertension, diabetes, stroke, and lung disease. The baseline characteristics of the included participants stratified by SES, depression status and cohorts were presented in S2-S4 Tables in S1 File.

Independent analysis of SES and depression on incident arthritis were illustrated in Table 2. After adjusting for covariates, individuals with medium education level had an HR of 1.12 (95% CI: 1.06, 1.17), while those with low

**Table 1. Baseline characteristics of included participants.**

| | Overall (N = 68,445) | No-arthritis (N = 53,969) | Arthritis (N = 14,476) | P value |
|---|---|---|---|---|
| **Age at baseline, mean (SD)** | **64.3 (9.7)** | **64.1 (9.7)** | **64.9 (9.4)** | **<0.001** |
| **Gender, N (%)** | | | | |
| Female | 34947 (51.1) | 26106 (48.4) | 8841 (61.1) | <0.001 |
| Male | 33498 (48.9) | 27863 (51.6) | 5635 (38.9) | |
| **Body mass index, N (%)** | | | | |
| Normal | 24493 (35.8) | 19784 (36.7) | 4709 (32.5) | <0.001 |
| Underweight | 2725 (4.0) | 2232 (4.1) | 493 (3.4) | |
| Overweight | 25556 (37.3) | 20288 (37.6) | 5268 (36.4) | |
| Obesity | 15671 (22.9) | 11665 (21.6) | 4006 (27.7) | |
| **Marital status, N (%)** | | | | |
| Married/partnered | 51506 (75.3) | 40996 (76.0) | 10510 (72.6) | <0.001 |
| Other | 16939 (24.7) | 12973 (24.0) | 3966 (27.4) | |
| **Ever smoked, N (%)** | | | | |
| No | 35855 (52.4) | 27938 (51.8) | 7917 (54.7) | <0.001 |
| Yes | 32590 (47.6) | 26031 (48.2) | 6559 (45.3) | |
| **Ever drank, N (%)** | | | | |
| No | 19086 (27.9) | 15158 (28.1) | 3928 (27.1) | 0.024 |
| Yes | 49359 (72.1) | 38811 (71.9) | 10548 (72.9) | |
| **[a]Physical activity, N (%)** | | | | |
| Less than once a week of moderate or vigorous physical activity | 20194 (29.5) | 16472 (30.5) | 3722 (25.7) | <0.001 |
| At least once a week of moderate or vigorous physical activity | 48251 (70.5) | 37497 (69.5) | 10754 (74.3) | |
| **Education, N (%)** | | | | |
| Low | 31881 (46.6) | 25103 (46.5) | 6778 (46.8) | <0.001 |
| Medium | 23868 (34.9) | 18581 (34.4) | 5287 (36.5) | |
| High | 12696 (18.5) | 10285 (19.1) | 2411 (16.7) | |
| **[b]Total household wealth, N (%)** | | | | |
| Q1 | 17119 (25.0) | 13170 (24.4) | 3949 (27.3) | <0.001 |
| Q2 | 17115 (25.0) | 13355 (24.7) | 3760 (26.0) | |
| Q3 | 17108 (25.0) | 13605 (25.2) | 3503 (24.2) | |
| Q4 | 17103 (25.0) | 13839 (25.6) | 3264 (22.5) | |
| **Socioeconomic status, N (%)** | | | | |
| High | 5653 (8.3) | 4648 (8.6) | 1005 (6.9) | <0.001 |
| Higher-middle | 23378 (34.2) | 18649 (34.6) | 4729 (32.7) | |
| Lower-middle | 29357 (42.9) | 22890 (42.4) | 6467 (44.7) | |
| Low | 10057 (14.7) | 7782 (14.4) | 2275 (15.7) | |
| **Depression, N (%)** | | | | |
| No | 48693 (71.1) | 39032 (72.3) | 9661 (66.7) | <0.001 |
| Yes | 19752 (28.9) | 14937 (27.7) | 4815 (33.3) | |
| **Hypertension, N (%)** | | | | |
| No | 41050 (60.0) | 33125 (61.4) | 7925 (54.7) | <0.001 |
| Yes | 27395 (40.0) | 20844 (38.6) | 6551 (45.3) | |
| **Diabetes, N (%)** | | | | |
| No | 58916 (86.1) | 46533 (86.2) | 12383 (85.5) | 0.037 |
| Yes | 9529 (13.9) | 7436 (13.8) | 2093 (14.5) | |
| **Stroke, N (%)** | | | | |
| No | 65587 (95.8) | 51765 (95.9) | 13822 (95.5) | 0.022 |
| Yes | 2858 (4.2) | 2204 (4.1) | 654 (4.5) | |

*(Continued)*

**Table 1.** (Continued)

|  | Overall (N = 68,445) | No-arthritis (N = 53,969) | Arthritis (N = 14,476) | P value |
|---|---|---|---|---|
| **Cancer, N (%)** |  |  |  |  |
| No | 64188 (93.8) | 50655 (93.9) | 13533 (93.5) | 0.102 |
| Yes | 4257 (6.2) | 3314 (6.1) | 943 (6.5) |  |
| **Lung disease, N (%)** |  |  |  |  |
| No | 64430 (94.1) | 50978 (94.5) | 13452 (92.9) | <0.001 |
| Yes | 4015 (5.9) | 2991 (5.5) | 1024 (7.1) |  |

Data were mean (SD) and n (%).

ᵃ For the MHAS cohort, physical activity was classified as "at least three times a week of moderate or vigorous physical activity" and "less than three times a week of moderate or vigorous physical activity".

ᵇ Total household wealth, Q means quartile by a country. For example, SHARE cohort includes multiple countries, we calculated quartile by each country, summed together in one cohort and finally pooled all cohorts. Q1-Q4: from low to high.

education level had an HR of 1.25 (95% CI: 1.19, 1.32) compared to the high education level. For wealth, compared to the highest quartile, the HRs were 1.08 (95% CI: 1.03, 1.13) for the third quartile, 1.16 (95% CI: 1.11, 1.22) for the second quartile, and 1.23 (95% CI: 1.17, 1.29) for the lowest quartile. Regarding SES, compared to the 'high' group, the HRs were 1.13 (95% CI: 1.05, 1.21) for 'higher-middle', 1.29 (95% CI: 1.21, 1.38) for 'lower-middle', and 1.36 (95% CI: 1.26, 1.47) for 'low'. Depression was associated with an increased risk of incident arthritis (HR 1.37, 95% CI: 1.32, 1.42).

Joint associations of socioeconomic factors and depression with incident arthritis were presented in Fig 1. There were significant interactions between socioeconomic indicators (education, wealth, SES) and depression status (p for interaction < 0.001). Irrespective of education, wealth, or SES levels, individuals with depression consistently exhibited the greater risk for incident arthritis. However, within each depression level, a graded inverse association was observed, where higher socioeconomic position was associated with decreased risk. These patterns were also found in joint effects of different combinations of education and wealth, and depression on incident arthritis in S2 Fig in S1 File. Additionally, in the analysis of the impact of socioeconomic factors on incident arthritis by depression status (Fig 2), it was found that lower levels of education, wealth, and SES were progressively associated with an increased risk of developing arthritis and these trends persisted among both depressed and non-depressed individuals. Dose-response associations between depression and incident arthritis by socioeconomic factors were shown in Fig 3. Dose-response curves, generated using restricted cubic splines, displayed similar and consistent non-linear patterns across different levels of socioeconomic factors (all P values for nonlinear < 0.05). Initially, the hazard ratios (HRs) were below 1, but as the depression scores increased, there was a continuous rise in HRs, highlighting clear positive associations.

Supplementary analysis (S3-S6 Figs in S1 File) found that joint effects of socioeconomic factors and depression on the development of arthritis were more pronounced in adults less than 60 years than those above 60 years and similar between females and males. Additionally, S5 Table in S1 File provided analyses of both the additive and multiplicative interactions between socioeconomic factors and depression with new-onset arthritis.

Sensitivity analyses, which involved rerunning models on sub-samples by excluding one cohort at a time (S6-S11 Tables in S1 File), showed only slight changes in the results. Furthermore, imputing missing values and treating missing values as a separate category did not significantly alter the effects (S13 Table in S1 File). Results from the random effects model, stratified Cox model, and multivariable Cox model remained largely consistent (S12 Table in S1 File).

**Table 2. Independent analysis of socioeconomic factors, depression on incident arthritis.**

| | Events | Total | Model1 HR (95%CI) | Model2 HR (95%CI) | Model3 HR (95%CI) |
|---|---|---|---|---|---|
| **Education** | | | | | |
| High | 2411 | 12696 | Ref. | Ref. | Ref. |
| Medium | 5287 | 23868 | 1.18 (1.13, 1.24) | 1.16 (1.10, 1.21) | 1.12 (1.06, 1.17) |
| Low | 6778 | 31881 | 1.50 (1.43, 1.58) | 1.33 (1.26, 1.40) | 1.25 (1.19, 1.32) |
| **Wealth** | | | | | |
| Fourth quartile | 3264 | 17103 | Ref. | Ref. | Ref. |
| Third quartile | 3503 | 17108 | 1.11 (1.06, 1.16) | 1.10 (1.04, 1.15) | 1.08 (1.03, 1.13) |
| Second quartile | 3760 | 17115 | 1.24 (1.18, 1.30) | 1.20 (1.15, 1.26) | 1.16 (1.11, 1.22) |
| First quartile | 3949 | 17119 | 1.35 (1.29, 1.42) | 1.28 (1.22, 1.34) | 1.23 (1.17, 1.29) |
| **Socioeconomic status** | | | | | |
| High | 1005 | 5653 | Ref. | Ref. | Ref. |
| Higher-middle | 4729 | 23378 | 1.21 (1.13, 1.30) | 1.17 (1.09, 1.25) | 1.13 (1.05, 1.21) |
| Lower-middle | 6467 | 29357 | 1.47 (1.37, 1.57) | 1.38 (1.29, 1.47) | 1.29 (1.21, 1.38) |
| Low | 2275 | 10057 | 1.69 (1.57, 1.82) | 1.47 (1.36, 1.59) | 1.36 (1.26, 1.47) |
| **Depression** | | | | | |
| No | 9661 | 48693 | Ref. | Ref. | Ref. |
| Yes | 4815 | 19752 | 1.51 (1.46, 1.56) | 1.41 (1.36, 1.46) | 1.37 (1.32, 1.42) |

Model1 was unadjusted (country as a strata term). Model2 adjusted for age at baseline, gender. Model3 further adjusted for body mass index, marital status, smoking, drinking, physical activity, prevalent hypertension, diabetes, stroke, cancer and lung disease. HR, hazard ratios; CI, confidence interval.

## Discussion

To our best of knowledge, this study is the first cross-cultural and longitudinal study to investigate independent and joint effects of SES, depression on incident arthritis in adults aged 50 and above. Our study reveals the intricate link between socioeconomic status, depression, and arthritis, advocating for targeted interventions to promote healthy aging. It underscores the importance of comprehensive public health strategies that tackle both socioeconomic and psychological factors to reduce arthritis globally.

Whereas low SES has been found to be associated with worse clinical outcomes and diminished functional capacity in arthritis patients [9,35,36], the association between SES and the risk of arthritis onset remains less understood. Over the past two decades, only a few case-control studies have investigated the association between SES and the risk of developing arthritis [13,14]. Consistent with our results, these studies found that low SES was positive associated with the development of arthritis. However, the strength of these findings is constrained by small sample sizes, potential recall and selection bias, and challenges in establishing temporal relationships. Our analysis revealed a clear trend: the risk of developing arthritis increased as levels of education and wealth decreased, a pattern more evident in the comprehensive SES score. This association held true across cohorts and remained significant even after adjusting for a range of demographic, lifestyle, and medical factors. For education, lower attainment often co-occurs with constraints in material resources, occupation, residential context, and access to care, including limited income and wealth, physically demanding or insecure work, neighborhood deprivation [37–39]. These conditions can plausibly increase the likelihood of incident arthritis by delaying preventive and early care, increasing biomechanical load and injury risk, constraining diet and physical activity, and sustaining chronic psychosocial stress and low-grade inflammation [14,38,40]. For wealth, financial resources play a critical role in access to healthcare across the continuum from prevention to advanced treatment. Insufficient resources

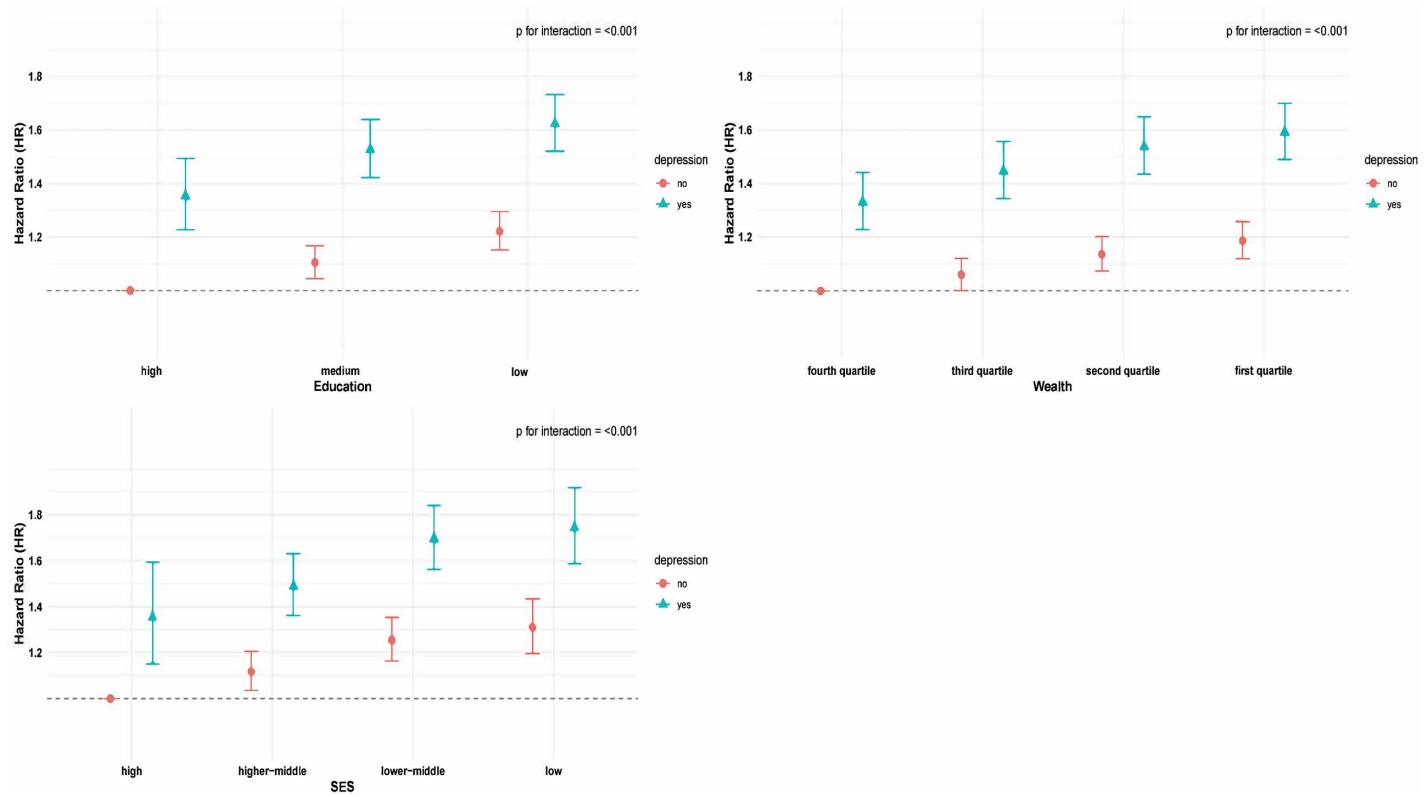

**Fig 1. Joint associations of socioeconomic factors and depression with incident arthritis.** All models were adjusted for age at baseline, gender, body mass index, country (strata), marital status, smoking, drinking, physical activity, prevalent hypertension, diabetes, stroke, cancer and lung disease. HR, hazard ratios; CI, confidence interval.

are associated with deferred or foregone care for chronic degenerative conditions, including constraints related to costs, the time required to obtain care, and travel distance. Lower wealth is also linked to adverse living conditions such as poorer nutrition and greater exposure to environmental risk factors, which can further predispose individuals to arthritis [39,40]. Taken together, lower SES amplifies the challenges posed by limited education and financial strain, substantially increasing the risk of developing arthritis.

Previous studies have established that depression could increase the risk of developing arthritis [15,16], and researchers have even uncovered a bidirectional relationship between the two conditions [41,42]. In this study, we found that depression was independently and significantly associated with increased risk of incident arthritis. The dose-response relationship analysis revealed that as depression scores increased, the risk of developing new-onset arthritis escalated across the entire socioeconomic spectrum. This suggested that even subclinical levels of depressive symptoms could contribute to increased arthritis risk, underscoring the importance of early mental health interventions. Incorporating mental health assessments into regular health screenings is imperative, especially for those aged 50 and above, a population where depression often remains underdiagnosed [43]. It is crucial to integrate mental well-being into preventive healthcare strategies, emphasizing the need for prompt intervention in depressive symptoms to prevent chronic diseases effectively. Furthermore, ensuring that mental health services are universally accessible and equitable across all socioeconomic strata is essential for addressing the widespread impact of depression on health outcomes.

| Variables | Levels | Events | Total | HR(95% CI) | |
|---|---|---|---|---|---|
| **No–depression** | **Education** | | | | |
| | high | 1896 | 10509 | Ref. | |
| | medium | 3912 | 18646 | 1.09 (1.03 to 1.16) | |
| | low | 3853 | 19538 | 1.21 (1.14 to 1.28) | |
| | **Wealth** | | | | |
| | fourth quartile | 2438 | 13328 | Ref. | |
| | third quartile | 2439 | 12566 | 1.05 (0.99 to 1.11) | |
| | second quartile | 2474 | 11970 | 1.12 (1.06 to 1.19) | |
| | first quartile | 2310 | 10829 | 1.17 (1.11 to 1.25) | |
| | **SES** | | | | |
| | high | 829 | 4869 | Ref. | |
| | high–middle | 3521 | 18143 | 1.11 (1.03 to 1.20) | |
| | low–middle | 4111 | 19930 | 1.24 (1.15 to 1.34) | |
| | low | 1200 | 5751 | 1.28 (1.17 to 1.41) | |
| **Depression** | **Education** | | | | |
| | high | 515 | 2187 | Ref. | |
| | medium | 1375 | 5222 | 1.14 (1.03 to 1.27) | |
| | low | 2925 | 12343 | 1.22 (1.10 to 1.35) | |
| | **Wealth** | | | | |
| | fourth quartile | 826 | 3775 | Ref. | |
| | third quartile | 1064 | 4542 | 1.09 (0.99 to 1.19) | |
| | second quartile | 1286 | 5145 | 1.16 (1.06 to 1.27) | |
| | first quartile | 1639 | 6290 | 1.21 (1.11 to 1.32) | |
| | **SES** | | | | |
| | high | 176 | 784 | Ref. | |
| | high–middle | 1208 | 5235 | 1.11 (0.95 to 1.30) | |
| | low–middle | 2356 | 9427 | 1.27 (1.09 to 1.48) | |
| | low | 1075 | 4306 | 1.32 (1.12 to 1.56) | |

0.5   1   1.5   2

**Fig 2. The effects of socioeconomic factors on incident arthritis by depression status.** All models were adjusted for age at baseline, gender, body mass index, country (strata), marital status, smoking, drinking, physical activity, prevalent hypertension, diabetes, stroke, cancer and lung disease. Ref., reference; HR, hazard ratios; CI, confidence interval.

However, how depression interacts with SES on the incidence of arthritis has yet to be explored. In this study, we found significant joint and synergistic effects between SES and depression on incident arthritis. The joint analysis revealed a pronounced gradient in the risk of developing arthritis based on the presence of depression and the level of socioeconomic factors, highlighting individuals grappling with psychological issues and socioeconomic disadvantages as particularly susceptible to arthritis. Individuals aged 50 and above, more prone to depression due to age-related psychological and physiological changes, were vulnerable to the compounded impact of these factors on arthritis risk. Therefore, it is imperative to focus more attention on these groups, advocating for timely and appropriate interventions to mitigate their heightened risk. The potential mechanisms driving these joint effects were multifaceted. Firstly, the physiological changes associated with aging, such as increased systemic inflammation and altered immune responses, can be exacerbated by depression [44,45]. This exacerbation can predispose older adults to arthritis by amplifying the inflammatory pathways linked to the disease's progression. Secondly, economic instability, limited access to healthcare, and reduced social

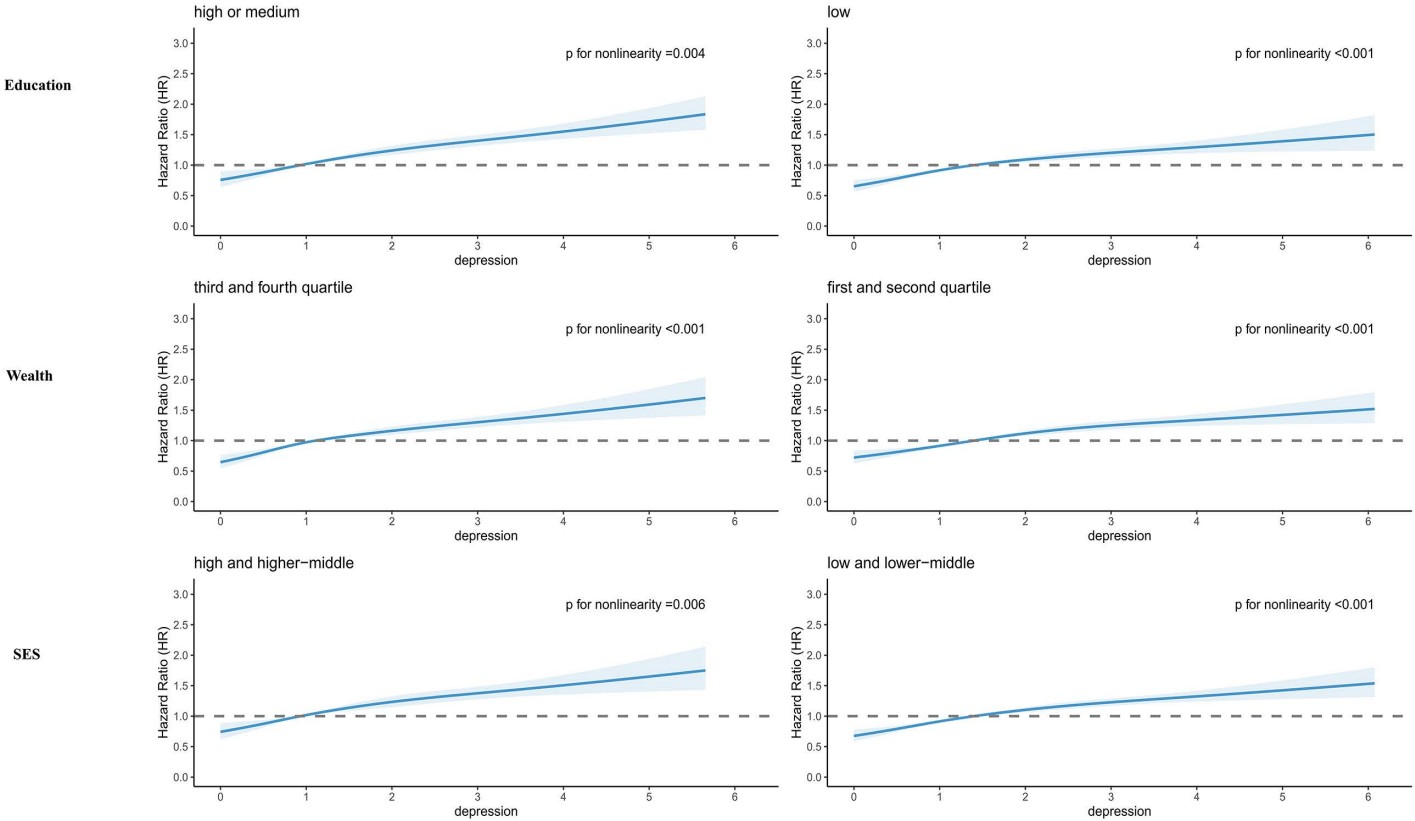

**Fig 3. Dose-response associations between depression and incident arthritis by socioeconomic factors.** To ensure comparability across cohorts with depression scales varying from 8 to 12 items, we standardized the score. Then, an adjustment was made by adding the absolute value of the minimum score, ensuring the minimum value started at zero. Restricted cubic splines were performed with four knots located at the 5th, 35th, 65th, and 95th percentiles. All models were adjusted for age at baseline, gender, body mass index, country (strata), marital status, smoking, drinking, physical activity, prevalent hypertension, diabetes, stroke, cancer and lung disease. HR, hazard ratios; CI, confidence interval.

support, which were more prevalent in population with lower SES, can exacerbate depression and its physiological consequences [46,47]. This can lead to a heightened state of chronic stress and inflammation, further increasing the risk of arthritis [48]. Lastly, the interaction between depression and SES may also be influenced by a lack of resources necessary for effective disease management and mental health care. A scarcity of accessible and age-appropriate health services could hinder effective management of depression and arthritis symptoms.

Notably, arthritis risks were higher among those with depression but decreased with increased socioeconomic advantages, a finding supported by joint and subgroup analyses. The findings suggested the protective role of education and financial benefits in mitigating health risks, demonstrating that an improvement in SES might decrease arthritis risk, even for those with depression. This emphasized the importance of socioeconomic factors in health management across all life stages, particularly in later years. While most individuals over 50 may be retired and their educational levels relatively fixed, improving SES still holds substantial value. At this life stage, enhancing SES might involve better living conditions, increased social participation, and improved access and quality of healthcare. Strategies such as boosting retirement benefits, providing community support, and improving healthcare services are key to elevating SES among the elderly. Additionally, ongoing health education is crucial to help them better manage depressive symptoms and efficiently utilize available resources and services for maintaining mental and physical health.

Given substantial cross-country differences in financing, benefits, service availability, workforce, geography, and social protection, implementation must be context specific. Countries with high patient cost sharing should reduce out of pocket expenses for core services and provide financial protection for essential diagnostics and medicines; systems with uneven geographic distribution of services should deploy mobile outreach, transport support, and tele-consultations; where primary care is underdeveloped, expanding standardized community protocols and establishing nurse led or physiotherapist led arthritis care pathways can improve timely treatment; where waiting times are long, increasing community rehabilitation capacity and introducing advanced practice roles can relieve bottlenecks. Aligning common clinical aims with country specific delivery offers an available path to earlier diagnosis, better symptom control, and more equitable outcomes for older adults, particularly those with lower education or limited wealth.

Our study possessed several strengths. Firstly, it ventured into a relatively unexplored domain by examining the joint effects of depression and SES on arthritis, thereby enriching the existing literature. The application of analyses to discern joint and synergistic effects offered a detailed view on how these factors interacted to influence arthritis risk, deepening the insights obtained. Secondly, our research underscored the significant role of mental health and SES in the context of arthritis, particularly among individuals aged 50 and above, indicating the potential of focused interventions to ameliorate mental and socioeconomic well-being in reducing arthritis risk. Lastly, the results of this cross-cultural, longitudinal study were generalizable because the large sample consisted of cohorts from across six studies, enhancing the depth and universality of our findings. This diversity allowed us to examine the relationship between depression and arthritis in a global context, shedding light on patterns and implications that transcended individual countries and hinted at more universally applicable insights.

Several limitations warranted consideration. First, the observational nature of the study hampered our ability to provide causal relationship. The consistent results in this study across different cohorts underscored the need for further research to explore the mechanisms underlying associations between SES, depression and incident arthritis. Second, despite our efforts to adjust for a wide range of covariates, the potential for unmeasured variables or residual confounding to have influenced the observed associations was an unavoidable limitation. Third, our dependence on self-reported data for assessing depression and arthritis diagnoses might have introduced reporting biases, potentially compromising the accuracy of these measures. Nonetheless, it is important to recognize that self-reports were commonly utilized in large-scale studies [49]. We suggest that future research could benefit from incorporating more objective diagnostic tools alongside self-reported data to enhance the reliability and validity of the findings. Fourth, the median follow-up duration of 5.9 years due to the data limitation. Confirmation in cohorts with longer observation windows is needed, as longer follow-up could modify both effect sizes and precision. Fifth, to preserve cross-country comparability, we operationalized SES using education and wealth only. Future work should incorporate additional SES dimensions related to individuals' quality of life and delineate how these conditions contribute to an increased risk of developing arthritis. Finally, our study measured SES and depression at a single point in time, which might not fully capture their potential fluctuations over the follow-up period.

In conclusion, our study sheds light on the intricate relationships between socioeconomic status, depression, and the incidence of arthritis, particularly in individuals aged 50 and above. The findings underscore the importance of considering both mental health and socioeconomic factors in arthritis prevention and management strategies. Such insights are invaluable for developing interventions that contribute to healthy aging.

## Supporting information

**S1 File. Supporting information**
(PDF)

## Author contributions

**Data curation:** Xiaomei Hu, Lingyan Duan.

**Formal analysis:** Rui Qing, Rongyan Mou, Shuqi Chen, Zhu Lan, Fengming Zou.

**Funding acquisition:** Xiaomei Hu.

**Investigation:** Lingyan Duan.

**Methodology:** Lingyan Duan.

**Project administration:** Xiaomei Hu.

**Resources:** Xiaomei Hu.

**Supervision:** Xiaomei Hu.

**Visualization:** Rui Qing, Rongyan Mou, Shuqi Chen, Zhu Lan, Fengming Zou.

**Writing – original draft:** Lingyan Duan, Rui Qing, Rongyan Mou, Shuqi Chen, Zhu Lan, Fengming Zou.

**Writing – review & editing:** Xiaomei Hu.

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
