## [Decision Letter · Decision Letter 0]

1 Aug 2025

Dear Dr. Hu,

Thank you for submitting your manuscript to PLOS ONE. After careful consideration, we feel that it has merit but does not fully meet PLOS ONE’s publication criteria as it currently stands. Therefore, we invite you to submit a revised version of the manuscript that addresses the points raised during the review process.

https://journals.plos.org/plosone/s/submission-guidelines#loc-laboratory-protocols . Additionally, PLOS ONE offers an option for publishing peer-reviewed Lab Protocol articles, which describe protocols hosted on protocols.io. Read more information on sharing protocols at https://plos.org/protocols?utm_medium=editorial-email&utm_source=authorletters&utm_campaign=protocols .

We look forward to receiving your revised manuscript.

Kind regards,

Modupe Akintomide

Academic Editor

PLOS ONE

**Journal Requirements:**

1. When submitting your revision, we need you to address these additional requirements. Please ensure that your manuscript meets PLOS ONE's style requirements, including those for file naming. The PLOS ONE style templates can be found at https://journals.plos.org/plosone/s/file?id=wjVg/PLOSOne_formatting_sample_main_body.pdf and https://journals.plos.org/plosone/s/file?id=ba62/PLOSOne_formatting_sample_title_authors_affiliations.pdf 2. We noticed you have some minor occurrence of overlapping text with the following previous publication(s), which needs to be addressed: https://linkinghub.elsevier.com/retrieve/pii/S2666756823000545https://www.thelancet.com/journals/lanhl/article/PIIS2666-7568(23)00195-2/fulltext? In your revision ensure you cite all your sources (including your own works), and quote or rephrase any duplicated text outside the methods section. Further consideration is dependent on these concerns being addressed. 3. Thank you for stating the following financial disclosure: Joint Science and Technology Funding Projects by Zunyi Bureau of Industry andScience and Technology and the Second Affiliated Hospital of Zunyi Medical University(HZ(2024)427)  Please state what role the funders took in the study.  If the funders had no role, please state: "The funders had no role in study design, data collection and analysis, decision to publish, or preparation of the manuscript." If this statement is not correct you must amend it as needed. Please include this amended Role of Funder statement in your cover letter; we will change the online submission form on your behalf. 4. Thank you for stating the following in the Acknowledgments Section of your manuscript: We thank all team members who have contributed to the HRS, ELSA, SHARE, KLoSA, CHARLS, and MHAS studies. This analysis used data or information from the Harmonized HRS, ELSA, SHARE, KLoSA, CHARLS, and MHAS datasets, and Codebooks developed by the Gateway to Global Aging Data, supported by the National Institute on Aging (grant numbers R01 AG030153, RC2 AG036619, and 1R03AG043052). For more information, please refer to https://g2aging.org. HRS is sponsored by the National Institute on Aging (grant number NIA U01AG009740) and is conducted by the University of Michigan. Funding for ELSA is provided by the National Institute of Aging (grants 2RO1AG7644–01A1 and 2RO1AG017644) and a consortium of UK Government departments coordinated by the Office for National Statistics. SHARE data collection has been funded by European Union (under grant agreement number 101102412) and the European Union’s Horizon 2020 research and innovation programme (under grant agreement numbers 870628 and 101015924), and various national funding sources gratefully acknowledged (https://share-eric.eu/infrastructure/funding). KLoSA is organised by the Korea Employment Information Service. CHARLS is supported by the Behavioral and Social Research division of the National Institute on Aging of the National Institute of Health (grants 1-R21-AG031372–01, 1-R01-AG037031–01, and 3-R01AG037031–03S1); the Natural Science Foundation of China (grants 70773002, 0910107022, and 71130002), the World Bank (contracts 7145915 and 7159234), and is conducted by Peking University. The Mexican Health and Aging Study is partly sponsored by the National Institutes of Health and National Institute on Aging (grant number NIH R01AG018016) in the USA and the Instituto Nacional de Estadísticay Geografía in Mexico. We note that you have provided funding information that is not currently declared in your Funding Statement. However, funding information should not appear in the Acknowledgments section or other areas of your manuscript. We will only publish funding information present in the Funding Statement section of the online submission form. Please remove any funding-related text from the manuscript and let us know how you would like to update your Funding Statement. Currently, your Funding Statement reads as follows: Joint Science and Technology Funding Projects by Zunyi Bureau of Industry andScience and Technology and the Second Affiliated Hospital of Zunyi Medical University(HZ(2024)427) Please include your amended statements within your cover letter; we will change the online submission form on your behalf. 5. Thank you for uploading your study's underlying data set. Unfortunately, the repository you have noted in your Data Availability statement does not qualify as an acceptable data repository according to PLOS's standards. At this time, please upload the minimal data set necessary to replicate your study's findings to a stable, public repository (such as figshare or Dryad) and provide us with the relevant URLs, DOIs, or accession numbers that may be used to access these data. For a list of recommended repositories and additional information on PLOS standards for data deposition, please see https://journals.plos.org/plosone/s/recommended-repositories. 6. If the reviewer comments include a recommendation to cite specific previously published works, please review and evaluate these publications to determine whether they are relevant and should be cited. There is no requirement to cite these works unless the editor has indicated otherwise. 

Reviewers' comments:

Reviewer's Responses to Questions

**Comments to the Author**

1. Is the manuscript technically sound, and do the data support the conclusions?

Reviewer #1: Yes

Reviewer #2: Yes

2. Has the statistical analysis been performed appropriately and rigorously?

Reviewer #1: Yes

Reviewer #2: Yes

3. Have the authors made all data underlying the findings in their manuscript fully available?

Reviewer #1: No

Reviewer #2: Yes

4. Is the manuscript presented in an intelligible fashion and written in standard English?

Reviewer #1: Yes

Reviewer #2: Yes

**Reviewer #1: ** Socioeconomic status, depression, and incident arthritis in adults aged 50 years and over: prospective evidence from 21 countries.

The main objective of the study is clearly presented, proposing a methodological approach that addresses an existent gap in the arthritis literature, investigating the effect of depression and/or socioeconomic status (SES) on incident arthritis

Introduction

The introduction highlights the importance of examining relationship between depression and SES in the incidence of arthritis, emphasizing the need to integrate these factors into the research in order to design strategies aimed at reducing the burden of this disease.

It is true that there Is a limited number of papers that refers to depression and SES as risk factor to incident arthritis but it is well known that at least depression has a bidirectional causal relationship with arthritis (i.e Brain Behav. 2024 Jun;14(6):e3551. Consequently authors should to elaborate more about depression or/and SES can be seen as a cause of incident arthritis.

Methods and Materials

This is a secondary analysis of six longitudinal aging studies. I think is more appropriate to refer to the six studies instead of the 21 countries.

The description of independent variables is clear enough but, please explain further if changes over time in these two variables were taken into account.

I consider adequate and pertinent the use of the various statistical methods and tests applied in the data analysis, particularly given that they were applied separately in every dataset by country(s), which allows to obtain more accurate and representative estimates tailored to each sample. In addition, it is relevant that the analysis was conducted both with and without missing data, enabling the assessment of how and to what extent the missing data may have influenced the results.

The specific ethical approvals for each cohort should be mentioned in this section. In the case of MHAS for example, the web page indicates what information is needed to include.

Results

The results are conclusive and present the values obtained from the applied tests and statistical models, both crude and adjusted, aimed at evaluating the extent to which various risk factors influence the development of arthritis in adults aged 50 years and older. The findings demonstrate that the study’s main objective addresses clearly and effectively explored. Specifically, depression is associated with an increased risk of developing arthritis. Therefore, significant associations are identified between several socioeconomic indicators and the incidence of the disease among participants in the longitudinal cohorts. In other words, individuals with an unfavourable socioeconomic status, low education attainment and depressive symptoms, tend to have higher risk of developing the disease. In contrast, participants with more favourable socioeconomic conditions, exhibited a lower risk, regardless of whether they reported symptoms of depression.

Discussion

The interpretation of the results is precise and valid, in the terms of the perspective of the study.

However, some issues should be considered in order to strengthen this section.

One limitation is the follow up. The variables relationship and the incidence could change if the follow up would be longer.

I also consider it essential to revisit this study from a class-based perspective. Although, the study identifies SES as a relevant variable, it does not delve into the multiple dimensions that a lower socioeconomic status entails in relation to individuals’ quality of life, nor how these conditions may contribute to an increased risk of developing arthritis.

The study adopts an approach that, while focusing on SES and depression, also centres on the level of education, under the assumption that lower educational attainment is associated with reduced knowledge of the disease and its symptoms, potentially leading to delays in diagnosis and appropriation treatment of arthritis. Moreover, it is important to emphasize that the lack of information about the disease is not exclusive to individuals with low levels of education; other fundamental aspects —such as economic access to medical services for treating chronic degenerative conditions, as well as issues related to time and distance— are also influenced by SES.

Another important aspect to highlight is that, although the data analysis by country helps to o avoid statistical bias, the proposed strategies and health policies are presented from a generalized perspective. This represents another limitation of the study, as solutions do not necessarily have to be, and often cannot be, homogenous.

Therefore, I argue that a more integral (holistic) approach that includes these diverse dimensions is essential for a truly effective approximation of the comprehension of arthritis in adults aged 50+.

**Reviewer #2:**  This study is written well, easy to understand and with academic language. Nice tables, graphs, figures. I do not have revision suggestions. The impact of this study across multiple populations is helpful because we practice medicine in a global environment. And additionally similar to the recent obesity study across populations then this multi-national study helps us to better understand that many health issues are shared issues thus we can begin trying to solve the globally instead of continuing to focus only on single populations. This study can be used to justify the inclusion of international participants in studies about arthritis and social determinants of health.

Methods are well described and Results indicate that the Methods were followed.

Results were presented in an understandable logical manner.

Discussion included the linkage to similar Results in other studies. Discussion included the limitations of the self reported data and self reported physician diagnosis.

Further work on identifying covariates and confounders was included.

**Do you want your identity to be public for this peer review?** For information about this choice, including consent withdrawal, please see our Privacy Policy

Reviewer #1: No

Reviewer #2: **Yes: ** Jeananne Elkins MPH, DPT, PhD

---

## [Author Response · Author response to Decision Letter 1]

16 Sep 2025

To the Editor, Reviewers, PLOS ONE

Date: September 2, 2025

Subject: Addressing Editor/Reviewer Comments for PONE-D-25-22108

Dear Editor and Reviewers,

We sincerely thank you for the constructive feedback provided by the editor and reviewers. We have carefully revised the manuscript and addressed all comments to the best of our ability. Kindly help us with your further concerns (if any) to improve our paper. Below we are providing our responses to the comments and actions that we took to address them. The responses are highlighted in blue.

Kind regards,

Authors of PONE-D-25-22108

Socioeconomic status, depression, and incident arthritis in adults aged 50 years and over: prospective evidence from 21 countries

PONE-D-25-22108

Socioeconomic status, depression, and incident arthritis in adults aged 50 years and over: prospective evidence from 21 countries

PLOS ONE

Dear Dr. Hu,

Thank you for submitting your manuscript to PLOS ONE. After careful consideration, we feel that it has merit but does not fully meet PLOS ONE’s publication criteria as it currently stands. Therefore, we invite you to submit a revised version of the manuscript that addresses the points raised during the review process.

Included

Included

Included

I If applicable, we recommend that you deposit your laboratory protocols in protocols.io to enhance the reproducibility of your results. Protocols.io assigns your protocol its own identifier (DOI) so that it can be cited independently in the future. For instructions see: https://journals.plos.org/plosone/s/submission-guidelines#loc-laboratory-protocols. Additionally, PLOS ONE offers an option for publishing peer-reviewed Lab Protocol articles, which describe protocols hosted on protocols.io. Read more information on sharing protocols at https://plos.org/protocols?utm_medium=editorial-email&utm_source=authorletters&utm_campaign=protocols.

We look forward to receiving your revised manuscript.

Kind regards,

Modupe Akintomide

Academic Editor

PLOS ONE

Journal Requirements:

Response:

We have checked the manuscript and filenames to ensure that they comply with PLOS ONE's style and naming requirements. We added “*” for corresponding author in the author list. We changed the manuscript to double-space paragraph format. Moreover, the format of all headings was also checked to meet the requirement.

https://linkinghub.elsevier.com/retrieve/pii/S2666756823000545

https://www.thelancet.com/journals/lanhl/article/PIIS2666-7568(23)00195-2/fulltext?

In your revision ensure you cite all your sources (including your own works), and quote or rephrase any duplicated text outside the methods section. Further consideration is dependent on these concerns being addressed.

Response:

We thank the editors for noting this issue. We carefully reviewed the two cited publications and our manuscript. We identified minor overlapping text in the Introduction and Discussion sections, and have now rephrased all such sentences to avoid duplication. We confirm that the revised manuscript now cites all relevant sources and no unacknowledged overlapping text remains.

Joint Science and Technology Funding Projects by Zunyi Bureau of Industry and

Science and Technology and the Second Affiliated Hospital of Zunyi Medical University(HZ(2024)427)

Response:

We appreciate the editor’s observation and have revised accordingly. We added the sentence “The funders had no role in study design, data collection and analysis, decision to publish, or preparation of the manuscript.” on the Funding section.

We thank all team members who have contributed to the HRS, ELSA, SHARE, KLoSA, CHARLS, and MHAS studies. This analysis used data or information from the Harmonized HRS, ELSA, SHARE, KLoSA, CHARLS, and MHAS datasets, and Codebooks developed by the Gateway to Global Aging Data, supported by the National Institute on Aging (grant numbers R01 AG030153, RC2 AG036619, and 1R03AG043052). For more information, please refer to https://g2aging.org. HRS is sponsored by the National Institute on Aging (grant number NIA U01AG009740) and is conducted by the University of Michigan. Funding for ELSA is provided by the National Institute of Aging (grants 2RO1AG7644–01A1 and 2RO1AG017644) and a consortium of UK Government departments coordinated by the Office for National Statistics. SHARE data collection has been funded by European Union (under grant agreement number 101102412) and the European Union’s Horizon 2020 research and innovation programme (under grant agreement numbers 870628 and 101015924), and various national funding sources gratefully acknowledged (https://share-eric.eu/infrastructure/funding). KLoSA is organised by the Korea Employment Information Service. CHARLS is supported by the Behavioral and Social Research division of the National Institute on Aging of the National Institute of Health (grants 1-R21-AG031372–01, 1-R01-AG037031–01, and 3-R01AG037031–03S1); the Natural Science Foundation of China (grants 70773002, 0910107022, and 71130002), the World Bank (contracts 7145915 and 7159234), and is conducted by Peking University. The Mexican Health and Aging Study is partly sponsored by the National Institutes of Health and National Institute on Aging (grant number NIH R01AG018016) in the USA and the Instituto Nacional de Estadísticay Geografía in Mexico.

Joint Science and Technology Funding Projects by Zunyi Bureau of Industry and

Science and Technology and the Second Affiliated Hospital of Zunyi Medical University

(HZ(2024)427)

Response:

We acknowledge this important point and made the suggested changes. We delete all funding-related text from the Acknowledgments section and add this to the cover letter and our Funding Statement remains unchanged. The revised Acknowledgments now read:

“Acknowledgments

We thank all team members who have contributed to the HRS, ELSA, SHARE, KLoSA, CHARLS, and MHAS studies.”

5. Thank you for uploading your study's underlying data set. Unfortunately, the repository you have noted in your Data Availability statement does not qualify as an acceptable data repository according to PLOS's standards.

Response:

Thanks for pointing it out. While we cannot directly upload the raw data to PLOS due to cohort-specific data use agreements, these datasets are fully de-identified and publicly accessible to qualified researchers via each study’s official repository. According to the other publications using the same cohorts, we changed our statement as follow:

“Data Availability

All data used in this study are drawn from six publicly available, harmonized longitudinal cohort studies within the Program on Global Aging, Health, and Policy. These datasets are not proprietary to the authors and are accessible to all qualified researchers through the official repositories listed below. All data are de-identified, and each study has established access procedures to protect participant confidentiality.

Health and Retirement Study (HRS), United States: sponsored by the National Institute on Aging and conducted by the University of Michigan. Public use datasets are available at https://hrs.isr.umich.edu/data-products.

Survey of Health, Ageing and Retirement in Europe (SHARE): coordinated by the Max Planck Institute. Public release datasets are available at https://share-project.org/data-access.html.

English Longitudinal Study of Ageing (ELSA), United Kingdom: funded by the National Institute on Aging and UK government departments. Data are available through the UK Data Service at https://discover.ukdataservice.ac.uk/series/?sn=200011.

Korean Longitudinal Study of Aging (KLoSA), South Korea: administered by the Korea Employment Information Service and approved by Statistics Korea. Data are available via the Korea Labor Institute at https://survey.keis.or.kr/eng/klosa/klosa01.jsp.

China Health and Retirement Longitudinal Study (CHARLS), China: approved by the Peking University Biomedical Ethics Review Committee. Data are available at https://charls.pku.edu.cn/en.

Mexican Health and Aging Study (MHAS), Mexico: sponsored by the Mexican National Institute of Statistics and Geography (INEGI), University of Texas Medical Branch, and the National Institute on Aging. Public release datasets are available at http://www.mhasweb.org”

Reviewers' comments:

Reviewer's Responses to Questions

Comments to the Author

1. Is the manuscript technically sound, and do the data support the conclusions?

Reviewer #1: Yes

Reviewer #2: Yes

2. Has the statistical analysis been performed appropriately and rigorously?

Reviewer #1: Yes

Reviewer #2: Yes

2. Have the authors made all data underlying the findings in their manuscript fully available?

Reviewer #1: No

Reviewer #2: Yes

Response:

We acknowledge this important point and made the suggested changes. However, all data underlying our analyses come from six harmonized, publicly available longitudinal cohort studies (HRS, SHARE, ELSA, KLoSA, CHARLS, and MHAS). These datasets are not proprietary to the authors; rather, they are made accessible to the international research community through well-established data access procedures that protect participant confidentiality. Researchers can freely register and download the de-identified data from each study’s official repository. In accordance with the PLOS data policy, we have clearly described the sources of data and provided links to the repositories in the Data Availability Statement. While we cannot upload the raw data directly to PLOS due to the terms of use of the cohort studies, the data are fully available to any qualified researcher without restriction beyond the standard registration process.

3. Is the manuscript presented in an intelligible fashion and written in standard English?

Reviewer #1: Yes

Reviewer #2: Yes

4. Review Comments to the Author

Reviewer #1: Socioeconomic status, depression, and incident arthritis in adults aged 50 years and over: prospective evidence from 21 countries.

The main objective of the study is clearly presented, proposing a methodological approach that addresses an existent gap in the arthritis literature, investigating the effect of depression and/or socioeconomic status (SES) on incident arthritis

Introduction

The intro

---

## [Editor Report · Decision Letter 1]

20 Oct 2025

Socioeconomic status, depression, and incident arthritis in adults aged 50 years and over: prospective evidence from six longitudinal aging studies

PONE-D-25-22108R1

Dear Dr. Hu,

We’re pleased to inform you that your manuscript has been judged scientifically suitable for publication and will be formally accepted for publication once it meets all outstanding technical requirements.

Kind regards,

Modupe Akintomide

Academic Editor

PLOS ONE

---

## [Editor Report · Acceptance letter]

PONE-D-25-22108R1

PLOS ONE

Dear Dr. Hu,

I'm pleased to inform you that your manuscript has been deemed suitable for publication in PLOS ONE. Congratulations! Your manuscript is now being handed over to our production team.

Kind regards,

on behalf of

Dr. Modupe Akintomide

Academic Editor

PLOS ONE